# Multiple-core-hole resonance spectroscopy with ultraintense X-ray pulses

Aljoscha Rörig [1,2,7], Sang-Kil Son [3,7] ✉, Tommaso Mazza[1], Philipp Schmidt [1], Thomas M. Baumann[1], Benjamin Erk [4], Markus Ilchen[1,4,5], Joakim Laksman [1], Valerija Music [1,4,5], Shashank Pathak[6], Daniel E. Rivas [1], Daniel Rolles [6], Svitozar Serkez [1], Sergey Usenko [1], Robin Santra [2,3], Michael Meyer [1] & Rebecca Boll [1] ✉

Understanding the interaction of intense, femtosecond X-ray pulses with heavy atoms is crucial for gaining insights into the structure and dynamics of matter. One key aspect of nonlinear light–matter interaction was, so far, not studied systematically at free-electron lasers–its dependence on the photon energy. Here, we use resonant ion spectroscopy to map out the transient electronic structures occurring during the complex charge-up pathways of xenon. Massively hollow atoms featuring up to six simultaneous core holes determine the spectra at specific photon energies and charge states. We also illustrate how different X-ray pulse parameters, which are usually intertwined, can be partially disentangled. The extraction of resonance spectra is facilitated by the possibility of working with a constant number of photons per X-ray pulse at all photon energies and the fact that the ion yields become independent of the peak fluence beyond a saturation point. Our study lays the groundwork for spectroscopic investigations of transient atomic species in exotic, multiple-core-hole states that have not been explored previously.

Extreme ultraviolet (XUV) and X-ray free-electron lasers (FELs) provide very intense pulses ($10^{13}$ photons per pulse) with ultrashort pulse durations (a few tens of femtoseconds) that enable the absorption of more than one photon per atom or molecule[1–3]. Such multiphoton interaction is a process of fundamental scientific interest because it enables studying the creation of (transient) ionic states of matter on timescales that were hitherto not accessible. In the X-ray regime, multiphoton inner-shell ionisation is dominated by sequential ionisation[4], while direct processes can become significant in the XUV regime[5–7]. This extreme regime of nonlinear photon–matter interaction is important for various applications, such as single-particle imaging of biological macromolecules[8–11], Coulomb explosion imaging[3,12,13], the production of warm dense matter[14], and the formation and control of plasmas in clusters or nanoparticles[15–17].

Far above inner-shell binding energies of a given atom and in the absence of saturation of the ionisation process, the yield of an ion charge state is proportional to $I^n$, with $I$ being the X-ray intensity (= number of photons per unit area and per unit time) and $n$ the average number of absorbed photons required to reach a given charge state[4]. In contrast to multiphoton or tunnel ionisation using optical and infrared laser pulses[18], the pulse duration was found to have a comparatively minor impact on the ion charge-state distributions in the X-ray regime[2,4], except for charge states created predominantly via double-core-hole states[2,19,20]. The X-ray fluence (= number of photons per unit area) was thus established as the most influential parameter for the resulting charge-state distributions. When the fluence becomes extremely high, the ion yields start to deviate from $I^n$, showing saturation. This saturation effect for ionisation is well known in the

[1]European XFEL, Schenefeld, Germany. [2]Department of Physics, Universität Hamburg, Hamburg, Germany. [3]Center for Free-Electron Laser Science CFEL, Deutsches Elektronen-Synchrotron DESY, Hamburg, Germany. [4]Deutsches Elektronen-Synchrotron DESY, Hamburg, Germany. [5]Institut für Physik und CINSaT, Universität Kassel, Kassel, Germany. [6]J. R. Macdonald Laboratory, Department of Physics, Kansas State University, Manhattan, KS, USA. [7]These authors contributed equally: Aljoscha Rörig, Sang-Kil Son. ✉e-mail: sangkil.son@cfel.de; rebecca.boll@xfel.eu

optical strong-field regime[18,21,22], and similar effects have previously been observed in FEL experiments performed in the XUV[1,23] and X-ray[24–26] regimes. However, to the best of our knowledge, the feature of saturation has not been exploited in high-intensity FEL applications yet. Here, we demonstrate how the deep saturation regime, in combination with a free tunability of the photon energy, facilitates ultra-high-intensity (transient) X-ray spectroscopy.

The photon-energy dependence of X-ray multiphoton absorption could, so far, only be investigated at very few selected photon energies due to the fixed-gap undulators used at previous FEL facilities and the corresponding need to completely re-tune the FEL at each photon energy. In contrast, several X-ray spectroscopy techniques are well-established in the single-photon absorption regime at synchrotron facilities, where wide-range energy scans are routinely performed[27]. While the photon-energy dependence was generally expected to map the decreasing photoabsorption cross section for increasing photon energy (far above any ionisation edges), the opposite trend has been reported for very high X-ray intensities[26], and transient resonances during charge-up[28] can lead to charge states significantly higher than expected at certain photon energies[24–26,29–31]. Transient resonances have recently been found to influence molecular multiphoton ionisation[32] and to dramatically enhance scattering cross sections in X-ray diffraction imaging[33], but they can also cause increased radiation damage[34]. These observations illustrate the complex interplay between imaging and transient electronic structure, as well as the necessity for a careful choice of all X-ray pulse parameters. Isolating the influence of individual parameters on the experimental results can be difficult, and volume integration[35] is usually required to make a quantitative comparison between theory and experiment. The need for taking into account a fluence distribution in the interaction volume often limits the degree of information that can be obtained in nonlinear experiments at high intensity because observables are usually governed by contributions from the lower intensity. Here, we demonstrate how this can largely be circumvented by exploiting saturation at very high fluences.

We present a joint theoretical and experimental study that sheds new light on the fundamental interaction of ultraintense soft-X-ray pulses with isolated atoms. The exceptionally high soft-X-ray pulse energies and the variable-gap undulators available at the Small Quantum Systems (SQS) instrument of the European X-ray Free-Electron Laser[36] facilitate the extraction of resonance spectra for all charge states up to aluminium-like xenon ($Xe^{41+}$) in the photon-energy range between 700 and 1700 eV. In certain photon-energy regions, the rich structures in the resonance spectra result from the dominance of previously postulated massively hollow atoms[37] created by multiple inner-shell photoabsorptions. Specifically, atoms featuring up to six simultaneous core holes are identified in the resonance spectra. There are a few experimental examples of double-core-hole electron spectroscopy for atoms[20] and molecules[38,39], but multiple-core-hole states have, to our knowledge, only been discussed in theory so far[37,40–42].

## Results

### Photon-energy-dependent charge-state distributions

Figure 1 shows measured (a) and calculated (b) xenon charge-state distributions (CSDs) between 700 and 1700 eV in steps of 25 eV. This photon-energy range covers the ground-state orbital binding energies of the $3s$, $3p$, and $3d$ subshells of xenon. Charge states between $Xe^{4+}$ and $Xe^{6+}$ result from single-photon absorption[43]. Furthermore, rich structures with several local maxima at significantly higher charge states are visible in both the experimental and calculated CSDs, which shift towards higher charge states as the photon energy increases. During the X-ray pulse, xenon atoms charge up in a sequence of one-photon absorptions and relaxation events, particularly Auger-Meitner decay cascades. Generation of the highest observed charge state, $Xe^{41+}$ at 1325 eV, requires absorption of more than 30 photons (see Supplementary Fig. S1).

In the sequential direct one-photon ionisation limit[24,31,37], shown by the dashed white line in Fig. 1b, the highest charge state $q$ is determined by the last ionic state $q-1$ that can be ionised with one photon from its electronic ground state. However, charge states significantly beyond this limit are observed in the CSDs. At specific photon energies and charge states, the increasing electron binding energies during the charge-up drive certain transitions between inner-shell and valence or Rydberg orbitals into resonance, thus leading to resonance-enabled or resonance-enhanced X-ray multiple ionisation (REXMI)[24,25].

The characteristic ion-yield maxima in Fig. 1, which shift as a function of photon energy, are a manifestation of the highly transient resonances. Relevant transition energies of the ground-state electronic configurations of charge state $q-1$ are indicated by coloured markers in both panels of Fig. 1. Changes in the slopes are related to the electronic configurations: The $O$ shell becomes empty at $Xe^{8+}$, the $N$ shell at $Xe^{26+}$, and the $3d$ subshell at $Xe^{36+}$. In principle, accessible resonant excitations from the $3p$ and $3d$ subshells can be expected to lead to an increased yield of certain charge states at given photon energies. However, while the ion yield maxima in Fig. 1 do indeed follow the general trend of the resonant transitions, it will become clear in the following that a correct assignment of the resonances requires considering the complicated multiple ionisation dynamics during the FEL pulse in detail.

To this end, we performed ab initio ionisation dynamics calculations using the XATOM toolkit[44,45]. We take into account volume integration in the calculations[35] (see Supplementary Discussion S1) because experimental ion spectra are always subject to a fluence distribution determined by the focused X-ray beam shape. As shown in Fig. 1b, the theoretical CSDs reproduce the overall features of the experimental ion-yield maxima well. However, there are some noticeable differences. The experimental CSDs show a marked void between the two ion-yield branches around 800–1000 eV, which is less pronounced in the theoretical CSDs, and a small shift of ~25 eV between experiment and theory, best seen between 700–800 eV. We attribute this to the finite accuracy of the electronic structure method used in XATOM (see Supplementary Table S1 for details). Moreover, the highest charge states observed in the experiment, around 1000–1400 eV, are missing in Fig. 1b. As a consequence, the ion yield piles up for the intermediate charge states ($Xe^{22+}$ to $Xe^{32+}$), thus enhancing structures related to the $3d \longrightarrow 6f$ excitation. We will see in the next subsection that the highest charge states can be reproduced by increasing the peak fluence in the calculation beyond the calibrated peak fluence (see Supplementary Fig. S2). On the other hand, the theoretical CSDs show a structure around $Xe^{22+}$ to $Xe^{26+}$ at 700–800 eV, which is absent in the experimental data. This can be attributed to a reduced peak fluence at these photon energies in the experiment [see Fig. 2c and the corresponding text].

### Resonance structures and peak-fluence dependence

Figure 2a–d show experimental (black and grey dashed lines) and theoretical (coloured lines) ion yields as a function of photon energy for four exemplary charge states, corresponding to lineouts of Fig. 1. These spectra display rich resonance structures which are absent in neutral xenon atoms[28] or are energetically inaccessible for low charge states. The most complex features are observed for the "intermediate" charge states, such as $Xe^{25+}$ [Fig. 2c]. The overall trends, particularly the positions of minima and maxima, are well reproduced by theory. A small shift, ~25 eV, of the theoretical ion-yield maxima towards higher photon energies with respect to the experiment [best seen in Fig. 2a] is caused by inaccuracies in the transition energy calculation (see Supplementary Table S1). Deviations between theory and experiment (e.g., for peaks F and K–N) will be discussed below.

Aside from the emergence of the resonance structures, one intriguing observation in Fig. 2 is the influence of varying the peak

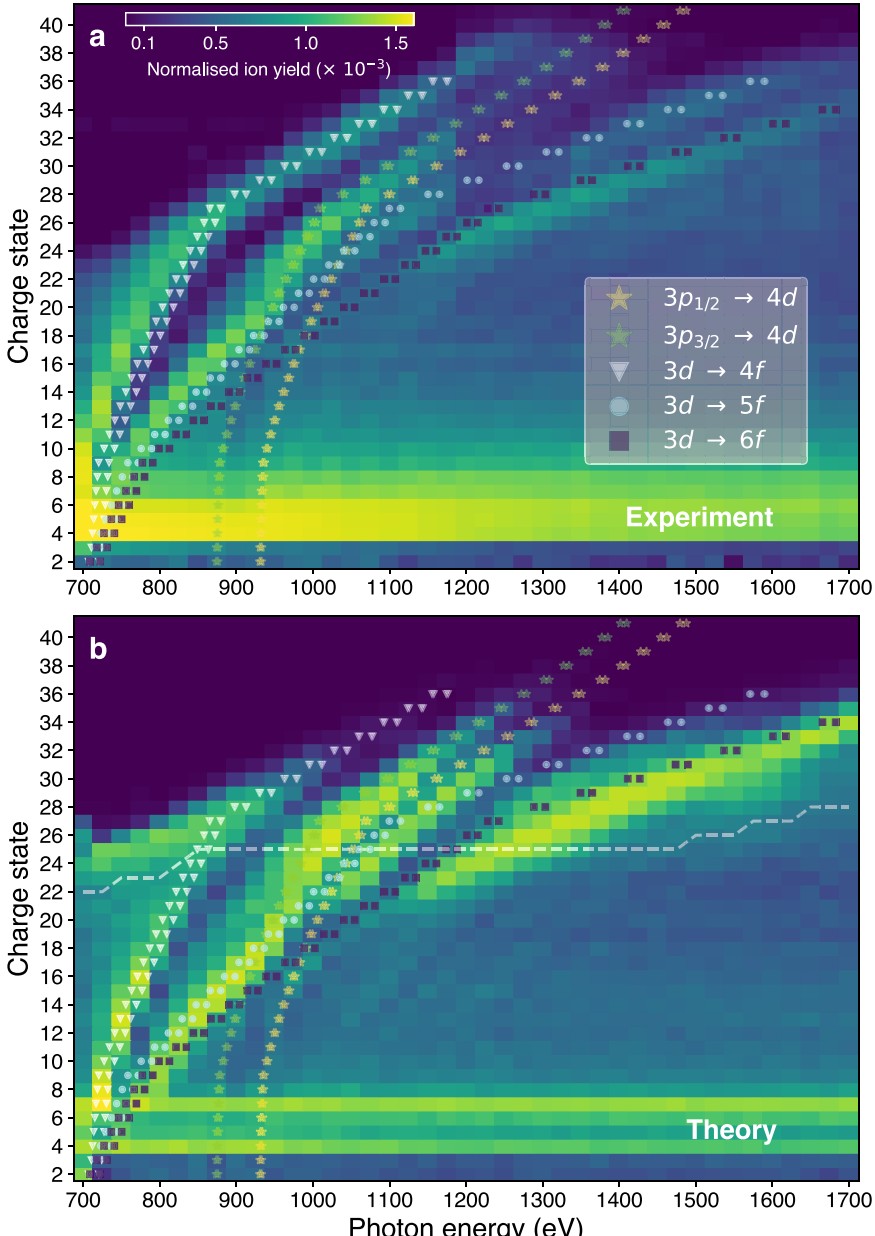

**Fig. 1 | Experimental (a) and calculated (b) xenon charge-state distributions as a function of photon energy.** Coloured markers show selected resonant transition energies in the ground-state configuration of each charge state $q$−1. (9.3 ± 0.2) × 10$^{12}$ photons per pulse on target were used in **a**. The theoretical results in **b** were volume-integrated[35] with a peak fluence of 1.2 × 10$^{12}$ photons/μm$^2$. The white dashed line in **b** indicates the sequential direct one-photon ionisation limit (see text). The ion yields are normalised to the sum of all detected (calculated) ion yields for all photon energies in the experimental (theoretical) data set. The discontinuity at 1200 eV in **a** is due to a change in the beamline optics (see Methods). The colour scale in **a** also applies to **b**.

fluence. Two experimental data sets at 50% (grey) and 100% (black) X-ray transmission are plotted. The peak fluence is generally assumed to be decisive for the resulting charge states in X-ray multiphoton ionisation. Indeed, for the high charge states, e.g., Xe$^{37+}$ [Fig. 2d], the ion yield increases nonlinearly for higher peak fluences. However, a different behaviour is observed for lower charge states. The resonance spectrum for Xe$^{15+}$ [Fig. 2a] is completely insensitive to the peak fluence in both experiment and theory within the chosen fluence range. For Xe$^{21+}$ and Xe$^{25+}$ [Fig. 2b, c], the spectra are identical for the higher photon energies but start to show variations as a function of fluence for low photon energies. Most noticeable is a strongly fluence-dependent peak F around 725 eV in the calculations. Its absence in the experimental data is consistent with a peak fluence of approximately 0.3 × 10$^{12}$ photons/μm$^2$ at this photon energy, matching our

fluence calibration [see Supplementary Fig. S3b]. Only peak G, which shows a weak sensitivity to the peak fluence in theory (in the chosen fluence range), is visible in the experimental data.

The peak-fluence dependence of peaks F, G, and K–N, in contrast to all other peaks, is attributed to a different ionisation mechanism. Further analysis of the theoretical data reveals that the number of photoionisation events exceeds the number of autoionisation events for these peaks, while Auger-Meitner-type autoionisation is the dominant mechanism to reach the final charge state for all other peaks (see Supplementary Fig. S4). In case a resonant excitation occurs in the outermost shell (for instance, peaks F and L, see Supplementary Fig. S5), only a few higher-lying (excited) electrons are available for relaxation, making autoionisation less likely. Peaks F, G, and K–N are the resonance structures corresponding to the highest charge states

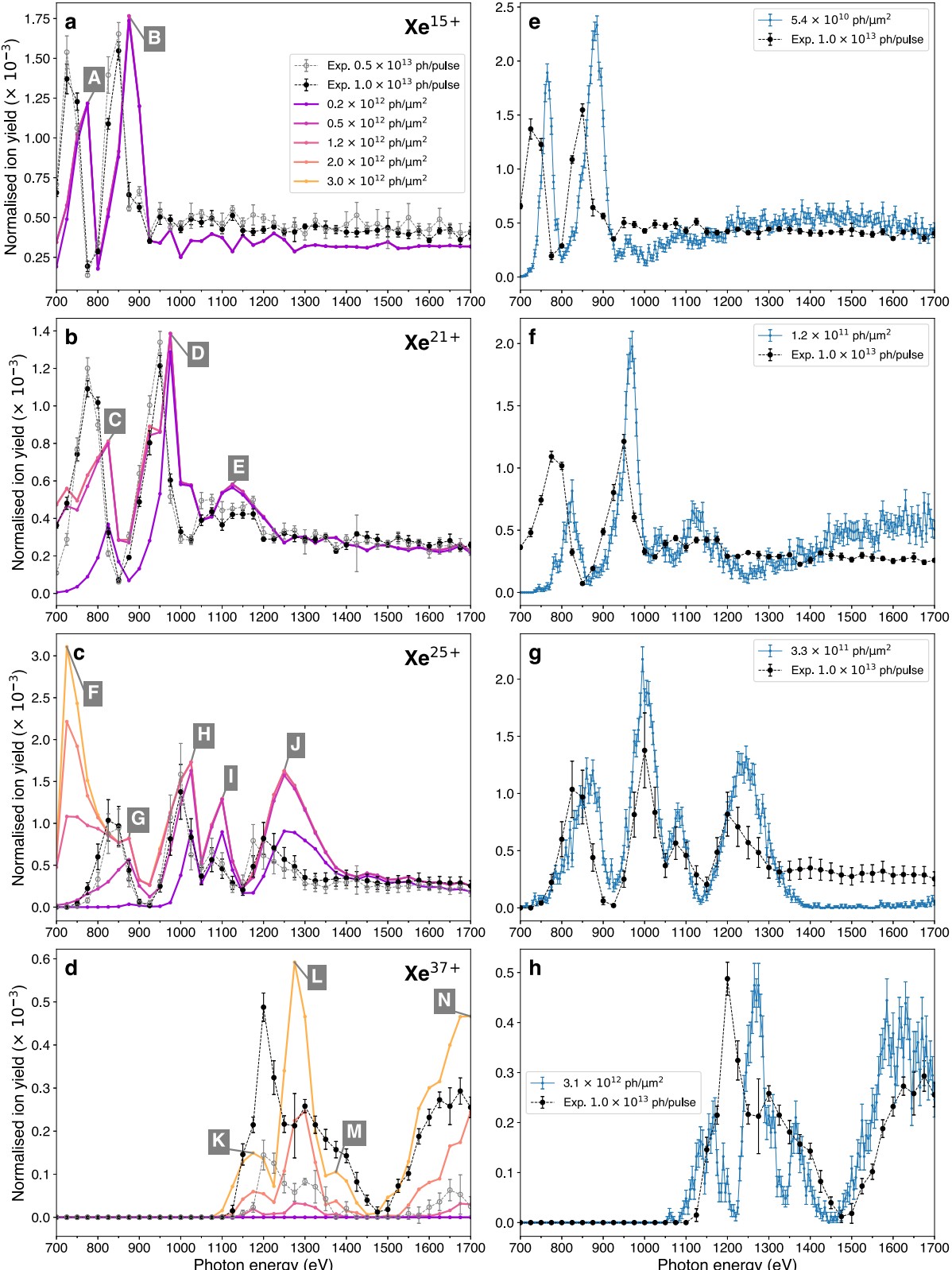

**Fig. 2 | Experimental and theoretical resonance spectra for four exemplary charge states of xenon.** The experimental data for $1 \times 10^{13}$ photons per pulse (black circles) in each panel correspond to horizontal lineouts of Fig. 1a. A second data set recorded at 50% X-ray transmission is also presented (grey open circles). **a**–**d** Volume-integrated calculations for different peak fluences (photons/$\mu m^2$) in different colours. They are identical in some cases, such that only one or a few lines are visible. Peaks are labelled A to N for further analysis in the main text and Supplementary Discussion S2. **e**–**h** Calculations without volume integration. Fixed

fluences (indicated in the legends) are chosen as an average of fluences that maximise the respective ion yield for a given range of photon energies. In these panels, a smaller step size of 5 eV is used in theory, and the theoretical data are normalised such that the sum of their ion yields equals the sum of the experimental ion yields in a respective panel. Theoretical error bars indicate the statistical uncertainty. The error bars of the experimental data include systematic and statistical errors (see Methods).

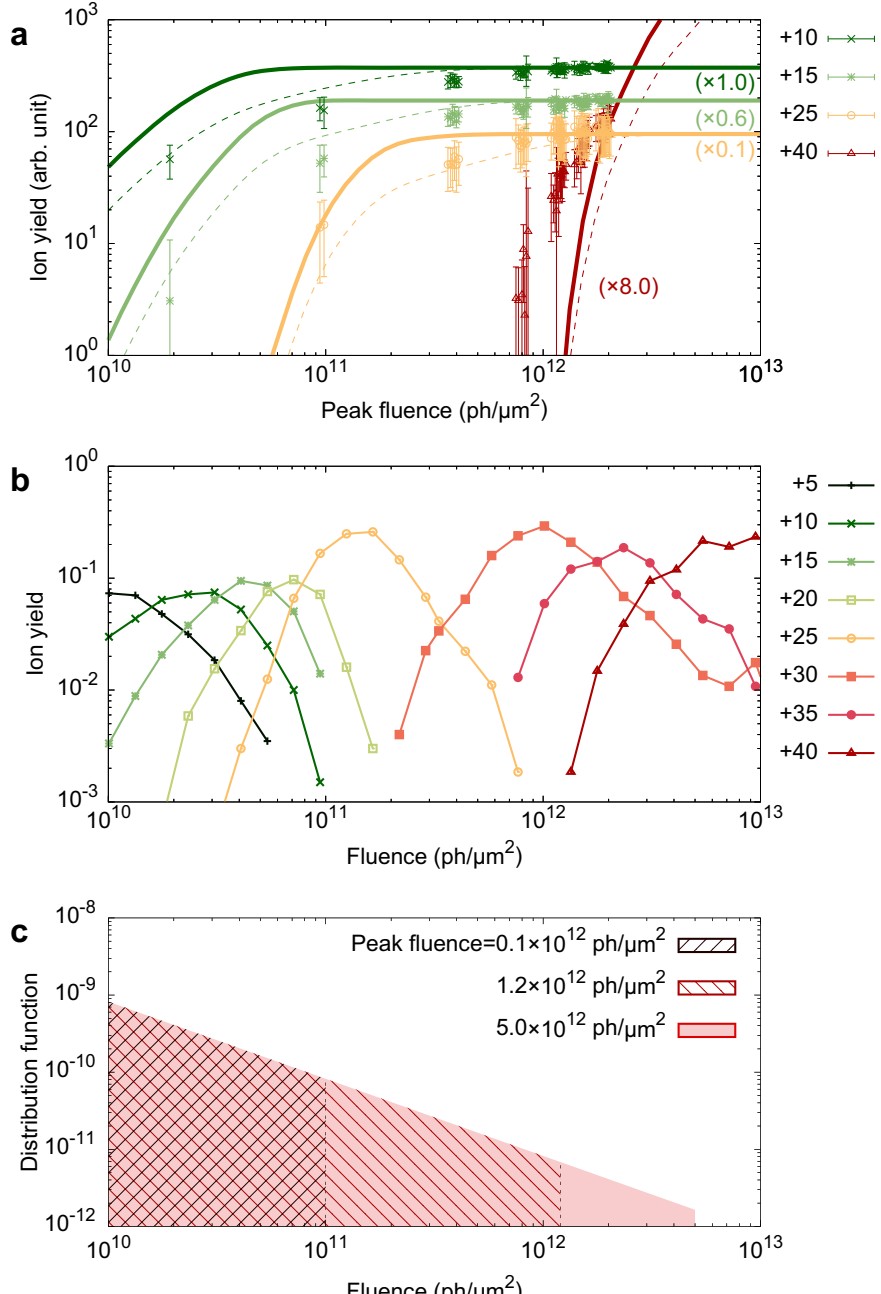

**Fig. 3 | (Peak-)fluence dependence of the ion yields of Xe at 1325 eV. a** Ion yields of Xe as a function of peak fluence. Theoretical ion yields are volume-integrated with a single (solid lines) or double (dashed lines) Gaussian spatial profile (see Supplementary Discussion S1 for details). Error bars of the experimental data include systematic and statistical errors (see Methods). The theoretical ion yields (both single and double Gaussian cases) are scaled to the experimental asymptotic values. The respective factors as specified next to the curves, with +10 being the reference. **b** Computed Xe ion yields as a function of fluence without volume integration. **c** Fluence distribution function in the X-ray focus for different peak fluences, assuming a single Gaussian spatial profile.

populated at the respective photon energy (see also Fig. 1). Those peaks have, therefore, not yet reached saturation and are thus sensitive to the X-ray fluence.

Figure 2a–d demonstrate that, beyond a certain saturation point, the overall resonance spectra for individual charge states are independent of the peak fluence used in volume integration. This means that volume integration is not necessary to capture essential features of the data. To further demonstrate this, we plot the corresponding resonance spectra calculated without volume integration in Fig. 2e–h. In view of this drastic simplification, the agreement of the calculated charge-state-specific resonance spectra with the experiment is still surprisingly good, illustrating that each charge state selects its own local fluence.

To further investigate the observed peak-fluence (in-)dependence of the resonance spectra, we plot the ion yields of several charge states of xenon at 1325 eV photon energy as a function of peak fluence in Fig. 3a. Experimental and theoretical ion yields (after volume integration) increase nonlinearly for lower peak fluences but become flat beyond a certain saturation peak fluence. The onset of this saturation starts later for higher charge states and is not yet reached for Xe$^{40+}$. We have multiplied the theoretical ion yields by individual factors (specified for each curve) to match the experimental data at the highest peak fluence because the absolute ion yields of intermediate charge states, such as Xe$^{25+}$, are overestimated, while high charge states such as Xe$^{40+}$ are underestimated. The latter is also reflected in Fig. 2d, where a peak

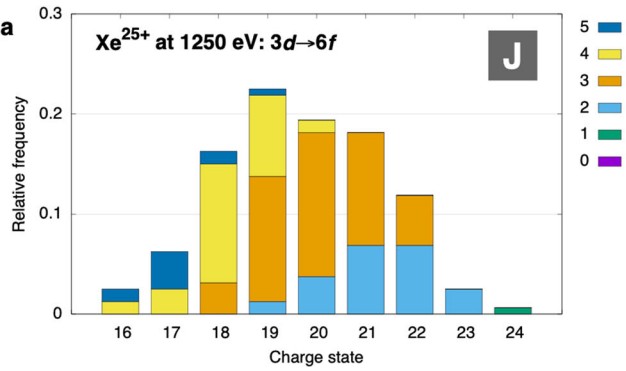

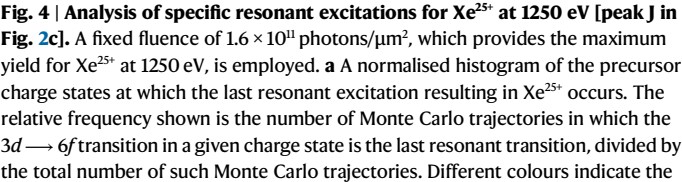

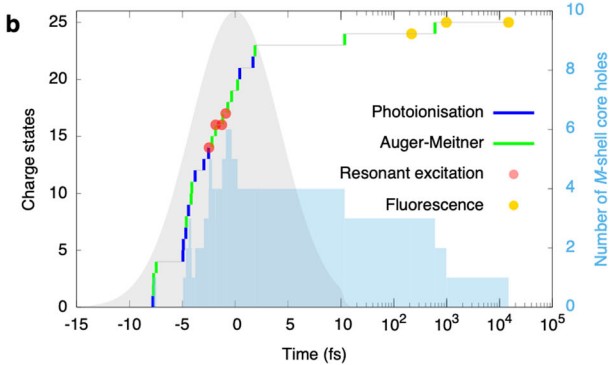

**Fig. 4 | Analysis of specific resonant excitations for Xe²⁵⁺ at 1250 eV [peak J in Fig. 2c].** A fixed fluence of $1.6 \times 10^{11}$ photons/μm², which provides the maximum yield for Xe²⁵⁺ at 1250 eV, is employed. **a** A normalised histogram of the precursor charge states at which the last resonant excitation resulting in Xe²⁵⁺ occurs. The relative frequency shown is the number of Monte Carlo trajectories in which the $3d \longrightarrow 6f$ transition in a given charge state is the last resonant transition, divided by the total number of such Monte Carlo trajectories. Different colours indicate the relative number of *M*-shell core holes present at the time of the last resonant excitation. **b** An exemplary ionisation pathway corresponding to peak J, specifically involving five core holes in the precursor charge state of +17. For the given ionisation pathway, the time evolution of the number of *M*-shell core holes (right *y* axis) is represented by the blue area. Note that the *x* axis is on a logarithmic scale after 10 fs.

fluence higher than the calibrated value is required to reproduce the experimental data of Xe³⁷⁺. Moreover, there are some inconsistencies in the saturation points with respect to the calibrated peak fluence. For instance, for peak C in Fig. 2b, the experimental data (100% and 50%) are almost the same (saturated), but the theoretical results at the calibrated values (corresponding to $0.4 \times 10^{12}$ photons/μm² and $0.2 \times 10^{12}$ photons/μm²) do not exhibit saturation. These points suggest that a careful validation of the atomic data employed in our calculation is required, especially for multiply excited ions in the soft-X-ray regime. The discrepancies between theory and experiment might be due to higher-order many-body processes[46,47], the chaoticity of self-amplified spontaneous emission (SASE) pulses[48], and coherence effects[28,49,50], which are not included in the model. The resonance positions are expected to profit from improved electronic structure theory[51], which affects resonant ionisation dynamics[52].

Without volume integration, as shown in Fig. 3b, we obtain the yield of a specific ion charge state as a function of fluence. This illustrates that every charge state is only generated in a relatively narrow fluence range $[f_1, f_2]$, which corresponds to a narrow radial range $[r_2, r_1]$ in a focused X-ray beam. For higher fluences, charge states are depleted because they become further ionised. Figure 3c shows a fluence distribution function for different peak fluences (different numbers of photons per pulse), assuming a fixed single Gaussian spatial profile. It becomes clear that an increase of the peak fluence does not change the abundance of lower fluences. The area $[= \pi(r_1^2 - r_2^2)]$ covering the fluence range $[f_1, f_2]$ in which a given charge state is generated remains constant when the number of photons per pulse increases[35,53], but the radii $r_1, r_2$ change depending on the peak fluence $F_0$: $r_i = \Delta \sqrt{(\ln\{F_0/f_i\})/(4 \ln 2)}$, where $\Delta$ indicates the focal size (full width at half maximum, FWHM). In combination, Fig. 3b, c explain why the volume-integrated ion yields in Fig. 3a become constant. Exceptions are high charge states, for example, Xe⁴⁰⁺, which are generated near the centre of the X-ray focus. The area in which they get created still increases when increasing the number of photons per pulse. Those charge states are not yet depleted in Fig. 3b, and saturation is not yet reached in Fig. 3a for the maximum peak fluence available in the experiment. An onset of saturation has also been observed in previous multiphoton absorption studies in the soft[24,25] and hard[26] X-ray regimes, but the peak fluence in those experiments was insufficient to reach a constant ion yield for many charge states.

In summary, this demonstrates that the resonance spectra in Fig. 2 are *independent* of the X-ray fluence and the focal beam shape as soon as the saturation regime is reached. This avoids the necessity of

volume integration in the calculations and makes multiphoton spectroscopy with ultraintense X-rays robust and insensitive, for example, to possible small variations of the X-ray focus size at different photon energies due to different beam divergence, and facilitates the unambiguous extraction of resonance features. The peak-fluence independence is a generic feature caused by the volume integration, and thus emerges for every sample and is independent of the target density used in the experiment. The results would be the same if only a single atom per pulse was located in the X-ray focal area.

## Resonance assignment and multiple-core-hole analysis

The peaks in the resonance spectra, such as those labelled A–N in Fig. 2, contain a plethora of information about the electronic structure and transient resonances. Based on the charge-state-dependent transition energies of certain ground-state resonances of charge state $q - 1$ (coloured markers in Fig. 1), we can tentatively assign peaks A and B in Xe¹⁵⁺ in Fig. 2a to the $3d \longrightarrow 4f$ and $3d \longrightarrow 5f$ transitions in Xe¹⁴⁺, respectively. However, this approach fails for the peaks in Xe²⁵⁺ in Fig. 2c—the ion yield maxima do not match any ground-state *M*-shell transition energies of Xe²⁴⁺ (see Supplementary Table S2). The photon energy at which a given peak in the resonance spectra occurs does not only depend on the charge state, but also on the number of core holes and the specific electron configuration, as illustrated in Supplementary Fig. S6. In the following, we illustrate how the correct assignment for Xe²⁵⁺ can be carried out.

In our calculations, X-ray multiphoton ionisation dynamics are described by a statistical ensemble of Monte Carlo trajectories in electronic configuration space[24,54]. First, we collected the calculated Monte Carlo trajectories ending with the given charge state for a fixed fluence, where the respective ion yield is maximised. In each trajectory, multiple resonant excitations can take place. The decisive transition for the structures in the resonance spectra of the final charge state is the last one. Therefore, we classified the Monte Carlo trajectories through electronic configuration space according to the last resonant transition, and the peak is then identified as the most probable excitation.

In Fig. 4, we illustrate the trajectory analysis for peak J as an example (see Supplementary Table S3 and Supplementary Discussion S2 for other peaks). At a fixed photon energy of 1250 eV, Fig. 4a shows a normalised histogram of the precursor charge states at which the last resonant excitation resulting in a final charge state of Xe²⁵⁺ occurs. The histogram shows a wide distribution of precursor charge states, demonstrating that the last resonant excitation almost never

happens at charge state $Xe^{24+}$, but can take place at charge states as low as $Xe^{16+}$. For this reason, an assignment of the resonance peaks is impossible without investigating the entire ionisation pathway with the help of ionisation dynamics calculations. After identifying the dominant transition, we find that the $3d \longrightarrow 6f$ transition causes the appearance of peak J.

Furthermore, the analysis of trajectories allows us to retrieve transient core-hole states which are populated during the charge-up. In Fig. 4a, colours indicate the number of core holes in the $M$ shell at the time of the last resonant excitation. Triple core holes are the dominant contribution, and up to quintuple core holes are detected. The last excitation creates one additional $M$-shell hole, i.e., sextuple core holes are present after the last excitation. Each core hole predominantly relaxes via Auger-Meitner decay, increasing the charge state by one. Therefore, lower precursor charge states exhibit the highest number of core holes; for example, four or five core holes at $Xe^{16+}$, which needs to gain nine more charges in order to be detected with a final charge state of +25. Excitations without core holes do not exist−peak J is exclusively produced by multiple-core-hole states. This is an impressive demonstration of the complexity of the transient electronic structure emerging upon the interaction of matter with ultraintense X-ray pulses.

Figure 4b shows one exemplary ionisation pathway among the Monte Carlo trajectories corresponding to peak J. It illustrates how 10 photoionisations, 4 resonant excitations, 15 Auger-Meitner decays, and 3 fluorescence decays lead to the final charge state $Xe^{25+}$. The blue area indicates the time evolution of the number of $M$-shell core holes, reaching a maximum of 6. All individual electron configurations and their lifetimes involved in this specific ionisation pathway are listed in Supplementary Table S4. Six core holes are formed at the peak of the X-ray pulse at $Xe^{17+}$ and $Xe^{18+}$; their lifetimes are 0.75 fs and 0.80 fs, respectively. As the charge state increases, Auger-Meitner-type ionisation channels become unfavourable and their lifetime becomes longer because only a few valence electrons remain. In the Monte Carlo trajectory shown, the quadruple-core-hole state ($3d_{3/2}^{-2}3d_{5/2}^{-2}$ from $Xe^{20+}$ to $Xe^{23+}$) survives until 12 fs and the triple-core-hole state ($3d_{3/2}^{-2}3d_{5/2}^{-1}$ at $Xe^{24+}$) lasts until 600 fs. The last autoionisation takes place at 600 fs and the remaining $M$-shell core holes are filled up via subsequent fluorescence events. The lifetimes of atomic species created via a sequence of core ionisation and excitation events vary by several orders of magnitude, depending on the charge state and the number of core holes. Thus, the multiple-core-hole resonance spectroscopy demonstrated in this work provides unique opportunities to examine exotic species with a wide range of lifetimes.

## Discussion

We have presented a type of resonance spectroscopy using ultraintense femtosecond X-ray radiation. Stable operation of the free-electron laser over an energy range from 700 to 1700 eV, in combination with advanced photon diagnostics and fast attenuation allowed us to maintain a constant number of photons per pulse on target during the scan. The resulting multiphoton resonance spectra unveil a wealth of structures, which can be assigned with the help of state-of-the-art theoretical calculations. We demonstrate that the resonance spectra become insensitive to the peak fluence above a certain saturation peak fluence, which allows isolating the effects of X-ray beam parameters that are otherwise intertwined with the dominant fluence dependence. In our case, this facilitates the characterisation of the transient resonant excitations during the charge-up, even without volume integration in calculations.

Transient multiple-core-hole states are found to be crucial for explaining the peaks in the resonance spectra, and some peaks are even exclusively formed via two or more core holes. This paves the way for multiple-core-hole (resonance) spectroscopy with ultraintense and ultrashort XFEL pulses. We demonstrate that extremely short-lived, as

well as unusually long-lived, highly charged ions in exotic electronic configurations can be created and probed simultaneously through interaction with intense soft-X-ray pulses. Such unusual atomic species may also be formed via collisions in outer space[55,56], which could be potential candidates for unidentified X-ray emission lines in astrophysics[57,58].

In principle, electron spectroscopy can provide additional information on the transient multiple-core-hole states. However, in practice, the interpretation would be hampered by many overlapping emission lines. Furthermore, at such high degrees of ionisation (up to 41 electrons emitted from a single atom), Coulomb interaction among the ejected electrons would inevitably broaden the emission lines. Ion spectroscopy is therefore advantageous for investigating multiphoton multiple ionisation dynamics at ultraintense X-ray fluences. High-resolution fluorescence measurements, while only representing a minority of all involved electronic transitions, will provide valuable complementary information to further benchmark theory in the future. Seeding[59,60] and two-colour schemes[61] may be exploited to systematically study the evolution of transient resonances both in the spectral and temporal domains. Furthermore, upcoming possibilities of tuning the pulse duration in addition to the photon energy will enable more detailed insights into transient electronic structure.

## Methods
### Experiment
The experiment was carried out at the SQS scientific instrument at the European XFEL[36]. Isolated xenon atoms were irradiated with ultraintense, femtosecond soft-X-ray pulses. Xenon gas was introduced into the Atomic-like Quantum Systems (AQS) experimental station through an effusive needle at an ambient pressure of around $3 \times 10^{-8}$ mbar. The ions resulting from the interaction with the X-ray pulses were recorded by a high-resolution ion time-of-flight (TOF) spectrometer[62] and fast analogue-to-digital converters with a resolution of 0.25 ns.

The photon energy was scanned over a range of 1 keV (700–1700 eV) in 25 eV steps while maintaining a constant high number of photons per pulse (0.5 or $1 \times 10^{13}$) throughout the scan. This was facilitated by (i) the tunable variable-gap SASE3 undulators[63], (ii) the high energies per pulse (2.3–6.4 mJ), and (iii) a fast-responding, 15-metre-long gas attenuator filled with nitrogen gas[64]. The attenuator was adjusted for every photon energy to compensate for the change in initial pulse energy and used to perform fluence scans over more than two orders of magnitude at fixed photon energies. An additional gas monitor detector[65] was installed downstream of the interaction chamber for this experiment, to characterise the number of photons on target in parallel, as described in Ref. 66. Peak fluences $> 1 \times 10^{12}$ photons/μm² were achieved due to the few-micron focus created by the SQS focusing optics[67] (see also Supplementary Fig. S3).

The calculated upper limit of the X-ray pulse duration was 25 fs based on the electron bunch charge of 250 pC in the accelerator. To date, no direct measurement of the X-ray pulse duration has been carried out at the European XFEL, but indirect measurements suggest that the pulse duration can be approximately 10 fs[68]. The European XFEL operated at a 10 Hz repetition rate, at which it provided bursts of electron pulses with an inter-pulse frequency of 2.25 MHz. We used every 32nd of these pulses to produce photons. The X-ray pulses thus had a spacing of 14.2 μs, chosen to avoid overlapping ion spectra. Overall, we received 250–350 X-ray pulses per second. The bandwidth was measured with a grating spectrometer in advance of the experiment and was 1–2% for all photon energies.

### Data analysis
In order to obtain quantitative charge-state distributions (CSDs), several analysis steps were carried out. Single-shot ion traces were obtained by cutting the recorded traces in 14.2 μs segments−the time

interval between FEL pulses in the burst. Subsequently, we applied a filter to only analyse FEL shots with pulse energies within one standard deviation of the average pulse-energy distribution of the pulse train. For each analysis step, a systematic error estimate is carried out, as detailed in the following. The statistical error is extracted by splitting the data set at each photon energy and peak fluence bin into four subsets and evaluating the deviation between the CSDs of the four subsets.

**Isotope deconvolution.** Up to $Xe^{15+}$, all seven stable isotopes of xenon were resolved without superimposing isotopes of higher charge states in the TOF spectrum. For charge states higher than $Xe^{15+}$, some peaks of higher mass and higher charge, $m_i/q_i$, start overlapping with those of lower mass and lower charge, $m_{i-1}/q_{i-1}$. We, therefore, applied a deconvolution algorithm to extract the ion yields of all charge states, using the known isotope distribution as an input. We use the convolution theorem by inverse Fourier transformation of the Fourier-transformed TOF signal (S) divided by the Fourier-transformed isotope structure (R) extracted from the averaged TOF signal. S and R are on a logarithmic scale before the Fourier transformation is applied, making the isotope peaks equidistant for all charge states within the isotope structure. Otherwise, the distances would scale with the inverse of the charge state $q$ and the deconvolution algorithm could not be applied. The algorithm's error is estimated using the difference between the input and output of the deconvolution procedure on a simulated xenon TOF spectrum.

**Charge-state-dependent detector efficiency.** The per-shot ion counts ($\approx 90$) were too high to safely neglect double counts in the same mass-over-charge peak. Thus, we did not attempt to convert the raw data to individual ion counts, e.g., via software constant fraction discrimination, but instead applied a charge-state-dependent correction factor to the ion yield of each charge state, accounting for the increase in average microchannel-plate (MCP) signal height for higher charge states[69,70]. This correction factor[71] was obtained by recording a separate calibration data set at significantly reduced gas pressure, for which constant fraction discrimination could be applied, and extracting the peak pulse height of the pulse-height distribution for each charge state. The error of the correction is estimated by the difference between the counted spectra and the corrected spectra of the calibration data set.

**Background subtraction.** Background signal from ionisation of residual gas mainly consists of oxygen ions from water. By evaluating the pulse-height distribution of xenon isotope 132 in counting mode, the oxygen contribution can be estimated through its distinctly lower MCP pulse heights in comparison to highly charged xenon ions with the same flight time. This fraction is subtracted from the ion yields of $Xe^{25+}$, $Xe^{33+}$ and $Xe^{41+}$, which overlap with $O^{3+}$, $O^{4+}$, and $O^{5+}$, respectively. The error estimation is based on the difference between spectra with and without background subtraction.

**Target density normalisation.** The two data sets above and below 1200 eV photon energy, as well as the data at 50% transmission, were recorded with slightly different gas pressures. Therefore, the ion yields were normalised to the gas pressure measured by an ion gauge in the interaction chamber parallel to the data recording.

## Modelling

To interpret the experimental data and to elucidate the underlying ionisation mechanisms, ab initio ionisation dynamics calculations using the XATOM toolkit[44,45] were performed. XATOM has recently been extended to incorporate resonance and relativistic effects[26,31]. For any given electronic configuration of Xe ions, the electronic structure was calculated on the basis of the Hartree-Fock-Slater method,

implementing first-order relativistic energy corrections. The atomic data, including photoabsorption cross sections, Auger-Meitner (including Coster-Kronig) rates, and fluorescence rates, were calculated in leading-order perturbation theory.

Using a rate-equation approach[2,48], the X-ray multiphoton ionisation dynamics[4] were simulated by solving a set of coupled rate equations with calculated atomic data. For the given range of photon energies, the ionisation dynamics of Xe are mainly initiated by $M$-shell ($n = 3$) ionisation. The number of coupled rate equations, which is equivalent to the number of electronic configurations that are formed by removing zero, one, or more electrons from initially occupied subshells ($n \geq 3$) of Xe ions and placing them into $(n, l)$-subshells ($n \leq 30$ and $l \leq 7$), is $\sim 4.2 \times 10^{60}$ (see Refs. [30,37]). To handle such an enormous number of rate equations, we used a Monte Carlo on-the-fly approach[54]. We assumed no contribution from direct (non-sequential) two-photon absorption[72] and no effect due to the chaoticity of FEL SASE pulses[48]. Higher-order many-body processes such as double photoionisation via shakeoff or knockout mechanisms[46] and double Auger-Meitner decay[47] were not included.

We used an energy bandwidth of 1% and a Gaussian temporal profile with a 10-fs FWHM in all our calculations. Unless otherwise noted, the theoretical results were volume-integrated[35] with a peak fluence of $1.2 \times 10^{12}$ photons/$\mu m^2$, which was obtained as the mean value of the calibrated peak fluences for 1200–1700 eV (see Supplementary Fig. S3), and a single Gaussian spatial profile was used in the volume integration. All X-ray beam parameters were kept constant while the photon energy was varied.

## Data availability

The raw data recorded for the experiment at the European XFEL are available at https://doi.org/10.22003/XFEL.EU-DATA-002310-00. The underlying data of the figures can be obtained from the corresponding authors upon request.

## Code availability

The data-analysis code is available from the corresponding authors upon request. An executable of XATOM is available at https://www.desy.de/~xraypac/.

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

## Acknowledgements

We acknowledge European XFEL in Schenefeld, Germany, for the provision of X-ray free-electron laser beam time at the SQS instrument and would like to thank the staff for their assistance. We also thank the operators and the run coordinators at DESY for their commitment and patience in tuning and performing the wide photon energy scans for the first time. We thank José R. Crespo López-Urrutia and Rui Jin for their help with FAC calculations. We thank Kai Tiedtke, Andrey Sorokin, Fini Jastrow and Yilmaz Bican for providing the gas monitor detector downstream of the instrument. We also thank Theophilos Maltezopoulos for his support. A.R. and M.M. acknowledge funding by the Deutsche Forschungsgemeinschaft (DFG, German Research Foundation)–SFB-925–project 170620586. M.I., V.M., and Ph.S. acknowledge funding from the Volkswagen foundation for a Peter Paul Ewald-fellowship. S.P. and D.R. were supported by the US Department of Energy, Office of Science, Office of Basic Energy Sciences, under contract no. DE-FG02-86ER13491.

## Author contributions

R.B. and S.-K.S. conceived the beam time. A.R., S.-K.S., T.M., P.S., T.M.B., B.E., M.I., J.L., V.M., S.P., D.E.R., D.R., S.S., S.U., M.M., and R.B. carried out the experiment. A.R., with help of P.S., T.M., and R.B., analysed the data. S.-K.S. carried out the calculations using the XATOM toolkit (developed by S.-K.S. and R.S.). A.R., R.B., S.-K.S., R.S., and M.M. interpreted the results and wrote the manuscript with input from all authors.

## Funding

## Competing interests

The authors declare no competing interests.
