## [Peer Review File · Nature Communications]

Multiple-core-hole resonance spectroscopy with ultraintense X-ray pulsesREVIEWER COMMENTS

Reviewer #1 (Remarks to the Author):

The paper by Rörig et al. reports a study on the interaction of intense, ultrashort X-ray pulses with heavy atoms. By using a X-ray free-electron laser that provides X-ray pulses over a wide photon energy range, they succeeded in mapping out the transient electronic structures for multiple ionization pathways. In addition, they demonstrated that the results obtained were independent of the peak laser fluence, making multiphoton spectroscopy with XFEL robust and insensitive to small variations of X-ray focus size, and they detected multiple-core-hole states that have not been explored to date.

The results are extremely interesting and the paper is well written. However, there were a few areas that were unclear. In order to make this paper suitable for publication, the authors should address the following points.

1. The CSDs in Fig. 1(b) are calculated for a peak fluence of 1.2×10^{12} ph/ μm^2 . However, the peak fluences are about 0.4×10^{12} and 1.2×10^{12} ph/ μm^2 in the experiment for photon energies below and above 1200 eV, respectively. Why were the calculations only performed for experimental peak fluence above 1200 eV? It would be interesting to see the calculation results for lower peak fluences.
2. In Fig. 1(b), the higher charge states observed in the experiment around 1000-1400 eV are missing in the calculation. The authors state that highly charged ion yields can be reproduced by increasing the peak fluence in the calculations. The absence of higher charge states in Fig. 1(b) should be discussed in more detail. Is the reason related to the assumptions that double photoionization, shake-off, and double Auger decay are not included in the calculations?
3. In the calculations, the authors assume that each pulse has a Gaussian temporal profile with a FWHM of 10 fs. However, the SASE FEL produces a train of pulses with random heights and durations. The 10-fs Gaussian profile corresponds to the average temporal profile for the pulses. The authors should address the validity of this assumption. Can the authors explain in more detail the influence of the intensity of individual SASE pulses on multiple ionization dynamics? Can the intensity spikes in the SASE pulses induce (non-sequential) two-photon absorption?
4. On Page 5, Line 105, the authors state that "For Xe²¹⁺ and Xe²⁵⁺ [Figs. 2(b) and (c)], the overall shapes of the resonance spectra remain unchanged for both experimental data sets, and the theoretical ion yields start to become sensitive to the peak fluence for photon energies lower than 975 and 1400

eV, respectively". Why are the experimental yields of Xe²¹⁺ and Xe²⁵⁺ insensitive to the peak fluence? For instance, the experimental Xe²¹⁺ spectrum shows a peak at 800 eV which is insensitive to the peak fluence. According to Fig. S2, the peak fluences in the experiment are 0.4×10^{12} and 0.2×10^{12} ph/ μm^2 at this photon energy. Under such conditions, the theoretical ion yields are strongly dependent on the peak fluence. Also, the experimental peaks at 850 and 1000 eV in the Xe²⁵⁺ spectrum are insensitive to the peak fluence, which is not consistent with the theory. The authors should address this large discrepancy between the theory and experiment.

5. In Fig. 3(a), theoretical ion yields are multiplied by individual scaling factors to match the experimental data. The authors should address these scaling factors not in the supplementary information but in the main text. In particular, it is highly recommended that the authors discuss in more detail the large difference between the theoretical and experimental yields for Xe⁴⁰⁺.

6. How was the peak fluence used in the analysis of specific resonance excitations in Figs. 4 and S4?

Reviewer #2 (Remarks to the Author):

The authors' study involves the interaction of Xe with high intensity X-ray pulses as a function of the photon energy. They discuss both experimentally and theoretically

at high fluence X-ray pulses that saturation is achieved. They also discuss how the formation of highly charged states is facilitated by transient resonances resulting from multiple core hole states. If the above is correct and I have correctly understood the main idea of the authors' work I find it quite interesting and worth publishing in Nature Communications assuming the following issues are addressed:

1) In the introduction, as it currently stands, it is not, in my opinion, crystal clear what the novelty of the authors' results is. Specifically, for instance, they mention that they perform for the first time a detailed study of the formation of charged states as a function of the photon energy. The novelty is the technical difficulty involved in performing the experimental studies that the authors have overcome? If so, it should be clearly stated and more references should be given to support the statement. Having just a sentence on this issue in the third paragraph is not sufficient. Moreover,

having a high number of core-holes with increasing photon energy is not surprising.

Can the authors clearly state in the introduction what main physical idea or mechanism does their study reveal? What is the previously unknown feature in transient resonances that the authors identify due to their detailed studies as a function of photon energy? In my opinion, the novelty in terms of physical mechanisms that the authors study allows does not come out in the introduction.

2) In the section Results, first paragraph, the authors state that "Charge states between Xe⁴⁺ and Xe⁶⁺ result from single photon absorption. Here, I would add a sentence that clearly explains what other processes are involved besides single-photon absorption in reaching from the ground state of Xe these charged states.

3) In the section "Resonance structures and peak-fluence dependence" the authors discuss Fig.2 and how the peak fluence changes or not the resonance structure as a function of photon energy. They state that theory well reproduces the overall trends observed in experiment. However, if I understand the figure correctly, while this is pretty much the case for Xe¹⁵⁺ and Xe²¹⁺ for the two higher Xe charged states this is not the case, for instance theory does not reproduce the correct location of the lower energy resonance around 800 eV for Xe²⁵⁺ and for Xe³⁷⁺ theory predicts higher yield for the resonance around 1300 eV rather than the one around 1200 eV. An explanation is necessary as to why theory is less accurate for these higher charged states.

4) For Fig. 4 the authors state in lines 158-159 "Lower precursor charge states exhibit the highest number of core holes" However they do not give an explanation as to why this is the case. Since one of the main points of the paper is the transient resonances facilitated by multiple core hole states, an explanation and more physical insight is necessary for the results presented in Fig. 4(a).

Reviewer #3 (Remarks to the Author):

The paper reports a combined experimental/computational work on the response

of Xenon atoms to intense X-ray SASE pulses of about 10 fs duration and a peak fluence of about 10^{10} to 10^{13} photons/ μm^2 in the regime of 700-1700 eV photons, generated by the European XFEL in Hamburg (SQS instrument).

As stated at the end of the abstract, the authors frame their work as

"Our study lays the groundwork for novel spectroscopies of transient atomic species in exotic, multiple-core-hole states that have not been explored previously." This is based on the observation that

"The extraction of resonance spectra is facilitated by the fact that the ion yields become independent of the peak fluence beyond a saturation point."

This fact is the basis to claim a new type of spectroscopy in the end of the second paragraph of the introduction: "Here, we demonstrate how the deep saturation regime, in combination with a free tunability of the photon energy, facilitates a new type of ultra-high-intensity (transient) X-ray spectroscopy."

The results reported are overall not surprising. Given the complexity of the physical processes involved, they are in general, in good agreement with the theoretical modeling using a Monte Carlo approach to cope with the enormous number of transition pathways calculated with rates based on a mean field description. This approach has proven to be useful in numerous XFEL related projects in the past.

A considerable part of the manuscript/supplement is devoted to analyze the properties of the photons from the XFEL beam which actually were interacting with the Xe atoms. This is challenging and important, as it is still very difficult to determine the properties of the SASE XFEL pulse to an extent that for a specific experiment, results can be transparently and clearly interpreted, in the words of the authors: "Isolating the influence of individual parameters

on the experimental results can be difficult, but in the saturation fluence regime, one can disentangle them to a certain extent."

To summarize, the punch line of the paper appears to be the experimental demonstration that beyond saturation the ion yield becomes independent of the photon fluence, rendering the interpretation of the results more reliable in the pertinent situation that the XFEL fluence cannot be accurately determined. The experimental achievement is a mode of operation for the XFEL with a constant number of photons reaching the target while the laser frequency is tunable (here from 700 to 1700 eV). This is achieved by an attenuator.

Does this constitute a "novel type of spectroscopy" ? And if so, what does it actually achieve ? Granted, this is a hitherto (to my knowledge) not possible mode of using XFEL pulses but to call this a novel type of spectroscopy one would have to demonstrate it to be realizable more universally with benefit also with other light sources.

It is unfortunate that experimental work at machines such as an XFEL must try to sell almost every campaign as "laying the groundwork" etc to achieve high impact publications which in turn seem to justify the enormous costs of beam time and the work of many scientists to realize such a experimental campaign.

Remaining unexplained discrepancies even after calibration of the intensity

(which is difficult, I understand) and focal averaging (also not easy), are

attributed to the usual suspects of the unpleasant characteristics of SASE XFEL pulses, in particular their spiky structure and more generally their partial coherence characteristics. There are well founded theoretical models for FEL generated partial coherence and in my opinion the authors would much more advance XFEL based science if they would use their atomic target and the now tunable XFEL mode to investigate and understand in detail the effect of the pulse characteristics on the observed ion yields

(thereby partially disentangling the effect of the pulse and possible limitations of the theoretical modeling).

Detailed remarks:

(a) I find the choice of data to displayed in figures well selected and illustrative. However, would it be possible to calculate more data points beyond the experimentally pre-set interval of 25 eV ? This would in particular help near energies of the "features" (A-F).

(b) The (dis)agreement of theory and experiment in Fig. 2 should be discussed and motivated in more detail: there is better agreement and less sensitivity to fluence for less charged ions, attributed to the regime of saturation. However, there are also significant shifts of the peaks, simply attributed to inaccuracies of the mean field theoretical description - can one get a bit more details here ? In particular to comment on the experimentally completely missing feature "C" as being "consistent" with lower fluence in the experiment without further quantification is not very satisfying.

In summary, the paper presents the results of an experiment with tunable XFEL radiation interacting with Xenon, analyzed with the XATOM program developed in Santra's group. This is good work and should be published.

To sell spectroscopy beyond the saturation limit as the novelty justifying publication in a high impact journal is, however, not convincing.

Most likely, the authors themselves are slightly split about this selling point of their paper since the heading does point in a (pleasantly) modest way to the full content of the paper.

As said before, what really would benefit the community and advance (SASE) XFEL science would be a careful assessment of the influence of the partially coherent light pulse on the experimental results on the occasion of a scenario which appears (compared to other setups) well controlled, with an atomic target (Xenon) and a good theoretical tool (XATOM and its extensions).

Reply to reviewer 1

Reviewer: *The paper by Rörig et al. reports a study on the interaction of intense, ultrashort X-ray pulses with heavy atoms. By using a X-ray free-electron laser that provides X-ray pulses over a wide photon energy range, they succeeded in mapping out the transient electronic structures for multiple ionization pathways. In addition, they demonstrated that the results obtained were independent of the peak laser fluence, making multiphoton spectroscopy with XFEL robust and insensitive to small variations of X-ray focus size, and they detected multiple-core-hole states that have not been explored to date.*

The results are extremely interesting and the paper is well written. However, there were a few areas that were unclear. In order to make this paper suitable for publication, the authors should address the following points.

Reply: We thank the reviewer for the positive assessment of our work. In the following, we address the unclear points and clarify the changes made in the revised manuscript.

Reviewer: *1. The CSDs in Fig. 1(b) are calculated for a peak fluence of $1.2 \cdot 10^{12}$ ph/ μm^2 . However, the peak fluences are about 0.4 and $1.2 \cdot 10^{12}$ ph/ μm^2 in the experiment for photon energies below and above 1200 eV, respectively. Why were the calculations only performed for experimental peak fluence above 1200 eV? It would be interesting to see the calculation results for lower peak fluences.*

Reply: Our aim was to demonstrate that the resonance spectra are no longer sensitive to the peak fluence once it has reached a saturation point. Therefore, we chose one consistent peak fluence for the entire photon-energy range in the calculations. We have added a new Supplementary Fig. S2 showing three different peak fluences. The fluence-independence of the intermediate charge states (Xe¹⁰⁺ to Xe²⁰⁺) is clearly visible, thus justifying the use of a single fluence in Fig. 1(b). The figure also addresses Point 2 below.

Reviewer: *2. In Fig. 1(b), the higher charge states observed in the experiment around 1000-1400 eV are missing in the calculation. The authors state that highly charged ion yields can be reproduced by increasing the peak fluence in the calculations. The absence of higher charge states in Fig. 1(b) should be discussed in more detail. Is the reason related to the assumptions that double photoionization, shake-off, and double Auger-Meitner decay are not included in the calculations?*

Reply: In the subsection “Resonance structures and peak-fluence dependence” of the Results section, we show how higher peak fluences can reproduce higher charge states in the experimental data, for example, Xe³⁷⁺ in Fig. 2(d). The newly-added Supplementary Fig. S2 (see also Point 1) further demonstrates that calculations with higher peak fluences generate the missing highest charge states for 1000–1400 eV, and we added the reference to Fig. S2 in the main text.

Regarding the remaining discrepancies between theory and experiment for high charge states, we suspect that several aspects are intertwined, including a lack of higher-order many-body processes (double photoionisation, shake-off, and double Auger decay), temporal shape (see

also Point 3), coherence effects, and inaccuracies of electronic structure theory employed in our ionisation dynamics model. However, a complete disentanglement of all aspects is far beyond the present study. We initially placed the corresponding discussion in the Supplementary Information but have now moved this text into the Results section, where we discuss the discrepancy between theory and experiment in Fig. 3(a) (see also Point 5).

Reviewer: 3. In the calculations, the authors assume that each pulse has a Gaussian temporal profile with a FWHM of 10 fs. However, the SASE FEL produces a train of pulses with random heights and durations. The 10-fs Gaussian profile corresponds to the average temporal profile for the pulses. The authors should address the validity of this assumption. Can the authors explain in more detail the influence of the intensity of individual SASE pulses on multiple ionization dynamics? Can the intensity spikes in the SASE pulses induce (non-sequential) two-photon absorption?

Reply: In the X-ray regime, where the energy of one photon is sufficiently high for ionisation, it is generally always possible to find an intermediate ionic state that can be reached by one-photon absorption. If a one-photon process is accessible, multiphoton processes occur predominantly via a sequence of one-photon processes rather than via direct multiphoton processes, even under high-intensity conditions [4,73]. Therefore, we did not include direct two-photon absorption in our ionisation dynamics model. A series of XFEL experiments so far support this sequential multiphoton ionisation model [2,24,25,26,73].

As the reviewer pointed out, the temporal shapes of SASE FEL pulses are fully chaotic. One might thus expect multiphoton ionisation processes to be enhanced due to the chaoticity. However, it has been demonstrated [48] that the statistical enhancement factor in X-ray multiphoton ionisation is actually much smaller compared to the enhancement in the optical domain and becomes close to one. This is particularly the case if the FEL coherence time (given by the duration of the SASE spike) is shorter than the lifetime of the intermediate state. Auger-Meitner lifetimes are in the order of a few fs, but can become shorter than 1 fs for the multiple-core-hole states in our case (see Table S4). The timescale of photoionisation can become as short as ~ 0.1 fs at the peak intensity. A realistic estimate of the coherence time of the FEL pulses is extremely difficult [see, e.g., <https://doi.org/10.23730/CYRSP-2018-001.539>], but a rough estimate from 1D FEL theory is in the 1- fs range near the saturation point for a cold electron beam [G. Geloni (European XFEL), private communication]. In conclusion, the coherence time of the X-ray pulses and the lifetimes of the intermediate states have a comparable order of magnitude.

In order to validate the assumption of 10- fs Gaussian pulses, we have calculated charge-state distributions (CSDs) for different temporal pulse shapes at a photon energy of 1325 eV and a fluence of $1.2 \cdot 10^{12}$ ph/ μm^2 (compare to Figs. 3 and 1(b)), see Figure R.1 in this letter. The flattop profile reflects the average of chaotic SASE pulses. We also generated 2000 SASE pulses with a bandwidth of 1%, following the procedure used in Ref. [48], and calculated the CSD for each of them. The average of this is shown as the cyan curve in Fig. R.1. The deviations to the result using Gaussian pulses are subtle. Thus, we do not expect the Gaussian shape to severely influence our calculated results. However, they might contribute to the missing highest charge states (see Point 2). We have thus included the chaoticity of SASE pulses as a potential candidate to improve the comparison between theory and experiment in the main text.

Figure R.1: Charge-state distributions of Xe at 1325 eV and $1.2 \cdot 10^{12}$ ph/ μm^2 calculated with three different temporal pulse profiles (without volume integration).

Reviewer: 4. On Page 5, Line 105, the authors state that “For Xe^{21+} and Xe^{25+} [Figs. 2(b) and (c)], the overall shapes of the resonance spectra remain unchanged for both experimental data sets, and the theoretical ion yields start to become sensitive to the peak fluence for photon energies lower than 975 and 1400 eV, respectively”. Why are the experimental yields of Xe^{21+} and Xe^{25+} insensitive to the peak fluence? For instance, the experimental Xe^{21+} spectrum shows a peak at 800 eV which is insensitive to the peak fluence. According to Fig. S2, the peak fluences in the experiment are 0.4 and $0.2 \cdot 10^{12}$ ph/ μm^2 at this photon energy. Under such conditions, the theoretical ion yields are strongly dependent on the peak fluence. Also, the experimental peaks at 850 and 1000 eV in the Xe^{25+} spectrum are insensitive to the peak fluence, which is not consistent with the theory. The authors should address this large discrepancy between the theory and experiment.

Reply:

The apparent discrepancy between experiment and theory at around 750 eV in Xe^{25+} was pointed out by all three reviewers (see Point 3 by Reviewer 2 and Point (b) by Reviewer 3), and we consequently rephrased the text describing Fig. 2 to provide a better explanation. It is important to note here that there are two different resonances (now labelled peaks F and G) between 700 eV and 900 eV in Xe^{25+} . While peak G dominates the experimental data, theory predicts peak F to become populated for peak fluences between $5 \cdot 10^{11}$ ph/ μm^2 and $1 \cdot 10^{12}$ ph/ μm^2 . As shown in Fig. S3, the experimental peak fluence in this photon-energy range was only about $3 \cdot 10^{11}$ ph/ μm^2 . Therefore, the fact that peak F was not populated in the experiment is consistent with the calibrated peak fluence.

Overall, we observe the peak-fluence independence after a saturation point for both theory and experiment in Fig. 2. However, the reviewer correctly noticed that the absolute peak fluence values at which the resonance spectra saturate are not exactly matched between experiment and theory everywhere in the spectra, for example, for peak C. Figure R.2 shows the same data

Figure R.2: Resonance profile of Xe^{21+} , but with additional calculated fluence points compared to Fig. 2(b).

as in Fig. 2(b) but with finer calculated peak-fluence points. The experimental data, recorded at the calibrated peak fluences of $4 \cdot 10^{11}$ ph/ μm^2 and $2 \cdot 10^{11}$ ph/ μm^2 (see Fig. S3), show fluence-sensitivity at the leftmost flank of the peak, at 700 and 725 eV, but are saturated for the remaining photon energies. The theory, however, predicts saturation of peak C only at about $4 \cdot 10^{11}$ ph/ μm^2 . Similarly, the onset of the saturation points for peaks G and H starts earlier in the experiment than in theory. These observations are probably related to the fact that the peak fluences theoretically needed to generate peaks K–N are well beyond the calibrated values (please see our reply to Point 2). In particular, the theoretical ion yields of high charge states are underestimated, reflected in the multiplication factors used in Fig. 3(a). We have added a sentence addressing the onset of the saturation points in the discussion of Fig. 3(a). We also note that assuming that the spatial profile is a single Gaussian and all other X-ray parameters are fixed, peak fluences that are required to reproduce the experimental data, particularly for high charge states, are roughly 4–10 times higher than the values resulting from the Ar calibration, as demonstrated in Fig. S2.

Reviewer: 5. In Fig. 3(a), theoretical ion yields are multiplied by individual scaling factors to match the experimental data. The authors should address these scaling factors not in the supplementary information but in the main text. In particular, it is highly recommended that the authors discuss in more detail the large difference between the theoretical and experimental yields for Xe^{40+} .

Reply: The scaling factors used in Fig. 3(a) were initially described in the Supplementary Information, but we have addressed this in the main text now. The under- or overestimation of the theoretical results compared to the experimental ion yields is closely related to the absence of the highest charge states discussed in Point 2 and the aspects described in Point 4. In the current study, we used the configuration-based XATOM toolkit, which is limited by the mean-field electronic structure theory. Resonant ionisation dynamics are particularly sensitive to

accurate transition energies [52]. Therefore, the resonance positions are expected to profit from an improved electronic structure theory, for instance, the state-resolved XATOM toolkit [51]. This improved toolkit has recently been applied to Ne atoms [52] but is currently far from a possible implementation of xenon due to the enormous complexity of the X-ray multiphoton ionisation dynamics.

Reviewer: 6. How was the peak fluence used in the analysis of specific resonance excitations in Figs. 4 and S4?

Reply: We used a fluence of $1.6 \cdot 10^{11}$ ph/ μm^2 without considering volume integration, which provides the maximum ion yield for Xe^{25+} ions at 1250 eV in our Monte Carlo simulations. This information has been added in the caption of Fig. 4. For the Monte Carlo analyses of other peaks shown in Fig. S5, we have specified a fluence value for each peak, listed in Table S3.

Reply to reviewer 2

Reviewer: *The authors' study involves the interaction of Xe with high intensity X-ray pulses as a function of the photon energy. They discuss both experimentally and theoretically at high fluence X-ray pulses that saturation is achieved. They also discuss how the formation of highly charged states is facilitated by transient resonances resulting from multiple core hole states. If the above is correct and I have correctly understood the main idea of the authors' work I find it quite interesting and worth publishing in Nature Communications assuming the following issues are addressed:*

Reply: We thank the reviewer for the overall positive assessment of our work and for raising a few important points that were not explained well enough in the first version. We have made a few notable changes to the manuscript, including adding new figures, and are convinced that this has improved the manuscript and addressed the unclear points.

Reviewer: *1) In the introduction, as it currently stands, it is not, in my opinion, crystal clear what the novelty of the authors' results is. Specifically, for instance, they mention that they perform for the first time a detailed study of the formation of charged states as a function of the photon energy. The novelty is the technical difficulty involved in performing the experimental studies that the authors have overcome? If so, it should be clearly stated and more references should be given to support the statement. Having just a sentence on this issue in the third paragraph is not sufficient.*

Moreover, having a high number of core-holes with increasing photon energy is not surprising.

Can the authors clearly state in the introduction what main physical idea or mechanism does their study reveal? What is the previously unknown feature in transient resonances that the authors identify due to their detailed studies as a function of photon energy? In my opinion, the novelty in terms of physical mechanisms that the authors study allows does not come out in the introduction.

Reply: The new technical possibility of scanning the photon energy over a wide range while keeping 10^{13} photons per pulse on target is certainly the enabling technology for the presented study, which we believe to be of significant interest for other nonlinear X-ray techniques in the future. This extreme intensity regime allowed us to reach saturation of many ion charge states. However, the pivotal scientific originality is the exploitation of the ion-yield saturation for spectroscopic purposes. We demonstrate that the saturation allows for surprisingly robust measurements of nonlinear phenomena, even though X-ray parameters such as the focus shape or the exact number of photons on target may not always be precisely known for experiments at SASE FELs.

We would like to clarify the role of the multiple core holes. Not an increase in photon energy per se, but the emergences of specific resonant excitations at certain photon energies and charge states are the facilitator for the emergence of multiple *simultaneous* core holes. We are not aware of any demonstration of more than two core holes in spectroscopic measurements in the existing literature. The surprising aspect, at least to us, was that some of the main features in our data, such as the peak J in Xe²⁵⁺, can only be rationalised through the population of

2–5 simultaneous transient core holes at charge states significantly lower than the final detected charge state. More conventional spectroscopy of highly charged ions, for example, using electron-beam ion traps, typically strip off valence electrons. In our case, the situation is different, as we study highly transient hollow atoms formed by inner-shell photoabsorptions with very short lifetimes. We have clarified these aspects in the Introduction.

Reviewer: 2) In the section *Results*, first paragraph, the authors state that “Charge states between Xe^{4+} and Xe^{6+} result from single photon absorption. Here, I would add a sentence that clearly explains what other processes are involved besides single-photon absorption in reaching from the ground state of Xe these charged states.

Reply: Initially, we listed those processes in the following paragraph, but we have slightly reworded this passage to explain it earlier on (see highlighted changes in the main text). The decisive aspect is that most of the final charge is generated by secondary processes, not by initial photoabsorptions. In addition, the new Fig. S6 shows the average numbers of photoabsorption and autoionisation processes for each individual resonance peak in Fig. 2. Autoionisation is the dominant channel for most peaks.

Reviewer: 3) In the section “Resonance structures and peak-fluence dependence” the authors discuss Fig.2 and how the peak fluence changes or not the resonance structure as a function of photon energy. They state that theory well reproduces the overall trends observed in experiment. However, if I understand the figure correctly, while this is pretty much the case for Xe^{15+} and Xe^{21+} for the two higher Xe charged states this is not the case, for instance theory does not reproduce the correct location of the lower energy resonance around 800 eV for Xe^{25+} and for Xe^{37+} theory predicts higher yield for the resonance around 1300 eV rather than the one around 1200 eV. An explanation is necessary as to why theory is less accurate for these higher charged states.

Reply: The apparent discrepancy between experiment and theory at around 750 eV in Xe^{25+} was pointed out by all three reviewers (see Point 4 by Reviewer 1 and Point (b) by Reviewer 3), and we consequently rephrased the text describing Fig. 2 to provide a better explanation. It is important to note here that there are two different resonances (now labelled peaks F and G) between 700 eV and 900 eV in Xe^{25+} . While peak G dominates the experimental data, theory predicts peak F to become populated for peak fluences between $5 \cdot 10^{11}$ ph/ μm^2 and $1 \cdot 10^{12}$ ph/ μm^2 . As shown in Fig. S3, the experimental peak fluence in this photon-energy range was only about $3 \cdot 10^{11}$ ph/ μm^2 . Therefore, the fact that peak F was not populated in the experiment is consistent with the calibrated peak fluence.

For the case of Xe^{37+} in Fig. 2(d), three different resonances contribute between 1100 and 1500 eV (now labelled K, L, and M), and a fourth is located at around 1700 eV (peak N). While the calculated energies of all four peaks in Xe^{37+} match the data well, we see a higher ion yield in peak L and a lower yield in peak K compared to the experiment. This observation is related to the observations in Fig. 3(a), where we had to multiply the theoretical curves with different scaling factors to match the experimental data (the scaling factors would be different for different photon energies). In addition, the new Fig. S6 demonstrates that peaks F, G, and K–N, which are sensitive to the peak fluence, are dominantly generated by photoionisation events rather than autoionisation events, which explains why they respond strongly to the number of

photons and, thus, the peak fluence. The remaining discrepancies, in particular for the highest charge states, are attributed to the following effects (see also our reply to Point 2 by Reviewer 1): lack of higher-order many-body processes, SASE chaoticity, coherence effects, and limitation of a mean-field description employed in ionisation dynamics calculations. We initially placed the discussion of these effects in the Supplementary Information but have now moved them into the main text.

Reviewer: 4) For Fig. 4 the authors state in lines 158-159 “Lower precursor charge states exhibit the highest number of core holes” However they do not give an explanation as to why this is the case. Since one of the main points of the paper is the transient resonances facilitated by multiple core hole states, an explanation and more physical insight is necessary for the results presented in Fig. 4(a).

Reply: A transient core hole at a given charge state will inevitably relax, predominantly by autoionisation via Auger-Meitner decay (see Fig. 4(b)), thus releasing an additional electron, which is the gist of the REXMI mechanism [24–26]. The histogram of the precursor charge states in Fig. 4(a) is a snapshot of the population of simultaneous core holes before reaching the final charge state of Xe^{25+} . Lower precursor charge states require several core holes to reach a given final charge state after their relaxation. We specified this point further in the manuscript.

Reply to reviewer 3

Reviewer: The paper reports a combined experimental/computational work on the response of Xenon atoms to intense X-ray SASE pulses of about 10 fs duration and a peak fluence of about 10^{10} to 10^{13} photons/ μm^2 in the regime of 700-1700 eV photons, generated by the European XFEL in Hamburg (SQS instrument).

As stated at the end of the abstract, the authors frame their work as “Our study lays the groundwork for novel spectroscopies of transient atomic species in exotic, multiple-core-hole states that have not been explored previously.” This is based on the observation that “The extraction of resonance spectra is facilitated by the fact that the ion yields become independent of the peak fluence beyond a saturation point.” This fact is the basis to claim a new type of spectroscopy in the end of the second paragraph of the introduction: “Here, we demonstrate how the deep saturation regime, in combination with a free tunability of the photon energy, facilitates a new type of ultra-high-intensity (transient) X-ray spectroscopy.”

The results reported are overall not surprising. Given the complexity of the physical processes involved, they are in general, in good agreement with the theoretical modeling using a Monte Carlo approach to cope with the enormous number of transition pathways calculated with rates based on a mean field description. This approach has proven to be useful in numerous XFEL related projects in the past.

A considerable part of the manuscript/supplement is devoted to analyze the properties of the photons from the XFEL beam which actually were interacting with the Xe atoms. This is challenging and important, as it is still very difficult to determine the properties of the SASE XFEL pulse to an extent that for a specific experiment, results can be transparently and clearly interpreted, in the words of the authors: “Isolating the influence of individual parameters on the experimental results can be difficult, but in the saturation fluence regime, one can disentangle them to a certain extent.”

To summarize, the punch line of the paper appears to be the experimental demonstration that beyond saturation the ion yield becomes independent of the photon fluence, rendering the interpretation of the results more reliable in the pertinent situation that the XFEL fluence cannot be accurately determined. The experimental achievement is a mode of operation for the XFEL with a constant number of photons reaching the target while the laser frequency is tunable (here from 700 to 1700 eV). This is achieved by an attenuator.

Does this constitute a “novel type of spectroscopy” ? And if so, what does it actually achieve ? Granted, this is a hithero (to my knowledge) not possible mode of using XFEL pulses but to call this a novel type of spectroscopy one would have to demonstrate it to be realizable more universally with benefit also with other light sources.

It is unfortunate that experimental work at machines such as an XFEL must try to sell almost every campaign as “laying the groundwork” etc to achieve high impact publications which in turn seem to justify the enormous costs of beam time and the work of many scientists to realize such a experimental campaign.

Remaining unexplained discrepancies even after calibration of the intensity (which is difficult, I understand) and focal averaging (also not easy), are attributed to the usual suspects of the unpleasant characteristics of SASE XFEL pulses, in particular their spiky structure

and more generally their partial coherence characteristics. There are well founded theoretical models for FEL generated partial coherence and in my opinion the authors would much more advance XFEL based science if they would use their atomic target and the now tunable XFEL mode to investigate and understand in detail the effect of the pulse characteristics on the observed ion yields (thereby partially disentangling the effect of the pulse and possible limitations of the theoretical modeling).

Reply:

We thank the reviewer for the detailed assessment of the expected impact of our manuscript. We do not share the general scepticism voiced by the reviewer for the following reasons.

Indeed, the employed Monte-Carlo approach was successfully applied several times before. The theoretical model itself is not the novelty in this work, though, but the context in which it is applied and the detailed conclusions which are drawn about transient resonances in highly charged ions featuring multiple simultaneous core holes. This was only possible due to the in-depth analysis of many individual calculated ionisation trajectories. Combined with the new experimental capability of tuning the photon energy, we could map out and assign transient resonances in hollow atoms for the first time.

The surprising aspect, at least to us, is that some of the main features in our data, such as the peak J in Xe^{25+} , can only be rationalised through the population of 2–5 simultaneous transient core holes at charge states significantly lower than the final detected charge state.

As the reviewer points out, the X-ray pulse parameters are often not well-characterised in experiments with SASE FELs, due to the very limited beamtime and unavailable, invasive, or very complicated diagnostics. While we devoted considerable effort to characterising our pulse parameters as much as possible, some uncertainties still remained. However, our subsequent analysis revealed that not all parameters are actually as important as previously expected, in particular, the exact X-ray fluence. The fact that the ion yields of many charge states become independent of the peak fluence for the very intense pulses allowed us to obtain resonance spectra insensitive to variations in the experimental peak fluence on the level shown in Fig. S3. Effectively, this means that the usual need for focal averaging or volume integration, as pointed out by the reviewer, can be completely eliminated in some cases. In order to emphasise this aspect further, we added Figs. 2(e)–(h) in the revised manuscript, showing calculated resonance spectra *without* volume integration. Even after this strong simplification, these show an impressive agreement with the experimental data. In addition, we added a corresponding clarification to the Introduction.

While we carried out a rather time-consuming tuning of the gas attenuator for each photon energy, it turned out that this would, in fact, not have been necessary, given the peak-fluence independence. The exceptionally high per-pulse intensity over the entire photon energy range is thus sufficient for conducting nonlinear X-ray spectroscopies in specific intensity regimes. Currently, the European XFEL is the only facility capable of this kind of spectroscopy, but several other facilities are being built or upgraded to provide comparable capabilities in the future.

We furthermore disagree with the assessment that: *"Remaining unexplained discrepancies [...] are attributed to the usual suspects of the unpleasant characteristics of SASE XFEL pulses, in particular their spiky structure and more generally their partial coherence characteristics."* This statement is not made anywhere in the manuscript and is incorrect. The remaining discrepancies

between theoretical and experimental results are attributed to a number of reasons, but the spiky temporal structure is certainly not the most significant. Please also refer to Figure R.1 in this letter and the reply to Reviewer 1.

We completely agree with the reviewer, though, that this new type of spectroscopy will be very valuable to understand pulse characteristics better in the future. In particular, we are interested in investigating the influence of the X-ray pulse duration, which is the most difficult parameter to characterise. In our last experiment, we could not measure, let alone change, the X-ray pulse duration. Yet, significant progress has since been made at the European XFEL to develop attosecond pulses, and we have already submitted a proposal for a corresponding follow-up beamtime.

Reviewer: *Detailed remarks:*

(a) I find the choice of data to displayed in figures well selected and illustrative. However, would it be possible to calculate more data points beyond the experimentally pre-set interval of 25 eV ? This would in particular help near energies of the "features" (A-F).

Reply: We thank the reviewer for this suggestion. We had chosen the theoretical data points to match the experimental energy steps, but of course, they can be calculated with an arbitrary increment. We added Figs. 2 as (e)–(h), including theoretical resonance spectra with 5-eV increments and without volume integration, thus showing the resonant features with higher resolution. These figures additionally demonstrate the comparatively minor impact of volume integration on the resonance spectra (see our response above).

Reviewer: *(b) The (dis)agreement of theory and experiment in Fig. 2 should be discussed and motivated in more detail: there is better agreement and less sensitivity to fluence for less charged ions, attributed to the regime of saturation. However, there are also significant shifts of the peaks, simply attributed to inaccuracies of the mean field theoretical description - can one get a bit more details here ? In particular to comment on the experimentally completely missing feature "C" as being "consistent" with lower fluence in the experiment without further quantification is not very satisfying.*

Reply: The apparent discrepancy between experiment and theory at around 750 eV in Xe^{25+} was pointed out by all three reviewers (see Point 4 by Reviewer 1 and Point 3 by Reviewer 2), and we consequently rephrased the text describing Fig. 2 to provide a better explanation. It is important to note here that there are two different resonances (now labelled peaks F and G) between 700 eV and 900 eV in Xe^{25+} . While peak G dominates the experimental data, theory predicts that peak F starts to become populated for peak fluences between $5 \cdot 10^{11}$ ph/ μm^2 and $1 \cdot 10^{12}$ ph/ μm^2 . As shown in Fig. S3, the experimental peak fluence in this photon-energy range was only about $3 \cdot 10^{11}$ ph/ μm^2 . Therefore, the fact that peak F (formerly labelled C) was not populated in the experiment is consistent with the calibrated peak fluence.

The energy shift of ~ 25 eV in Fig. 2(a) is attributed to the mean-field approach for calculating transitions involving the $3d$ orbital because both peaks A and B are assigned as the transition from $3d$ (Table S3), and those transitions show a systematic shift of ~ 23 eV in Table S1. Other shifts like C and E in Fig. 2(b) can be attributed to the same shift involving the $3d$ subshell because $3d \rightarrow 4f$ and $3d \rightarrow 6f/7f$, respectively, are involved (Table S3). For peak D, this is

not the case because it is assigned as $3p_{3/2} \rightarrow 4d$. According to the peak assignments given in Table S3, we suspect the shift near peak D is due to $3d \rightarrow 5f$ that can be mixed with other $3p$ transitions.

Reviewer: *In summary, the paper presents the results of an experiment with tunable XFEL radiation interacting with Xenon, analyzed with the XATOM program developed in Santra's group. This is good work and should be published. To sell spectroscopy beyond the saturation limit as the novelty justifying publication in a high impact journal is, however, not convincing. Most likely, the authors themselves are slightly split about this selling point of their paper since the heading does point in a (pleasantly) modest way to the full content of the paper.*

As said before, what really would benefit the community and advance (SASE) XFEL science would be a careful assessment of the influence of the partially coherent light pulse on the experimental results on the occasion of a scenario which appears (compared to other setups) well controlled, with an atomic target (Xenon) and a good theoretical tool (XATOM and its extensions).

Reply: We thank the reviewer for acknowledging our work's quality and recommending the publication of our results. We believe we emphasised the novel aspects of this work in our replies and with the corresponding changes to the introduction in the manuscript. While we do not expect the partial coherence to have the most significant impact, for the reasons stated above, we certainly plan to advance this technique to unravel further details of the interaction between heavy atoms and intense X-ray pulses. We are convinced that it will become valuable to better understand the pulse characteristics, in particular, the influence of the X-ray pulse duration, which will be freely tuneable in the future. A dedicated downstream diagnostics section, currently being built at the SQS instrument, will allow a shot-to-shot measurement of the X-ray temporal shape using angular streaking simultaneously to the ion spectra. In this way, it will be possible to study the influence of the pulse characteristics and potentially use the ion spectra themselves for a pulse-duration determination in the future. It would also be extremely interesting to make a complementary measurement at a seeded X-ray FEL with fully coherent pulses, but this currently only exists in the extreme ultraviolet range (FERMI in Trieste).

REVIEWERS' COMMENTS

Reviewer #1 (Remarks to the Author):

I thank the authors for their detailed response. The authors have included most of my suggestions in a convincing way. The revised manuscript is recommended for publication.

Reviewer #2 (Remarks to the Author):

Dear Editor,

I am satisfied with the changes made by the authors who I believe satisfactorily address the comments of all reviewers. I suggest publication to Nature Communications.

Reviewer #3 (Remarks to the Author):

The detailed response to (all) referees' comments are greatly appreciated.

The revisions have made the manuscript clearer, especially regarding the discrepancy of theory and experiment in Figure 2. The introduction has also improved to convey the main message (I very much agree with the second reviewer's comment, that the main message needs to fill with content - claiming "a novel kind of spectroscopy" does not say anything.)

The mode of XFEL operation (constant high number of photons/pulse over a wide energy range) is a step forward. *It should be mentioned in the abstract.* Also, the experimentally demonstrated fact, that one does not need to pay attention to volume integration and structure of the pulse beyond saturation is valuable. (Thank you for the theoretical illustration with figure R.1 in the reply.)

Since the two other reviewers are positive about publication in the journal chosen, I do not want to stay in that way. I would suggest, however, to include

the constant photon number mode in the abstract.

Reply to reviewer 1

Reviewer: I thank the authors for their detailed response. The authors have included most of my suggestions in a convincing way. The revised manuscript is recommended for publication.

Reply: We thank the reviewer for the positive feedback and the endorsement for publication at Nature Communications.

Reply to reviewer 2

Reviewer: Dear Editor, I am satisfied with the changes made by the authors who I believe satisfactorily address the comments of all reviewers. I suggest publication to Nature Communications.

Reply: We thank the reviewer for the positive feedback and the endorsement for publication at Nature Communications.

Reply to reviewer 3

Reviewer: The detailed response to (all) referees' comments are greatly appreciated. The revisions have made the manuscript clearer, especially regarding the discrepancy of theory and experiment in Figure 2. The introduction has also improved to convey the main message (I very much agree with the second reviewer's comment, that the main message needs to fill with content - claiming "a novel kind of spectroscopy" does not say anything.)

*The mode of XFEL operation (constant high number of photons/pulse over a wide energy range) is a step forward. *It should be mentioned in the abstract.* Also, the experimentally demonstrated fact, that one does not need to pay attention to volume integration and structure of the pulse beyond saturation is valuable. (Thank you for the theoretical illustration with figure R.1 in the reply.)*

Since the two other reviewers are positive about publication in the journal chosen, I do not want to stay in that way. I would suggest, however, to include the constant photon number mode in the abstract.

Reply:

We thank the reviewer for the positive feedback and support in publishing our manuscript at Nature Communications.

We have added the statement: "The extraction of resonance spectra is facilitated by the possibility of working with a constant number of photons per X-ray pulse at all photon energies and the fact that the ion yields become independent of the peak fluence beyond a saturation point." in the abstract to address the suggestion of including the constant number of photons in the abstract.